# 2EZBFT for Decentralized Oracle Consensus with Distant Smart Terminals

**DOI:** 10.3390/s25206268

**Published:** 2025-10-10

**Authors:** Yuke Cao, Kun She

**Affiliations:** School of Information and Software Engineering, University of Electronic Science and Technology of China, Chengdu 611731, China; 202222090603@std.uestc.edu.cn

**Keywords:** distributed oracle, data collection, consensus mechanism, blockchain technology

## Abstract

In geo-distributed deployments, sensor data are collected under the coordination of smart terminals and relayed on-chain via decentralized oracles. A motivating scenario involves healthcare networks where regional hospitals submit aggregated medical data to blockchain systems while maintaining strict information security—often designating one gateway per region for external communication. Long geographical distances between smart terminals stress traditional consensus with excessive network overhead and limited efficiency. To address this, we propose a layered BFT consensus method, 2-layer EaZy BFT (2EZBFT). The system forms multiple independent groups of smart terminals and builds a two-layer consensus architecture—“intra-group synchronization, inter-group consensus”—to complete cross-group data aggregation and final on-chain consensus. This layered design reduces intra-group communication complexity by lowering the number of nodes per group and reduces cross-group interactions via leader-side aggregation, thereby lowering overall network overhead. Compared with other BFT algorithms, the proposed scheme improves the efficiency of data collection and on-chain reporting while ensuring consensus security and consistency. Experiments show improvements in metrics such as network overhead and consensus latency. In a discrete-event simulation with an asymmetric WAN latency matrix and geo-partitioned groups, 2EZBFT achieves up to 45% higher throughput than flat BFT algorithms such as PBFT and HotStuff under high load. It provides a practical path for efficient data interaction in decentralized oracles and offers guidance for improving the performance of blockchain–real-world data exchange.

## 1. Introduction

Decentralized oracles enable smart contracts to react to real-world events by bridging on-chain logic with off-chain data sources. In many practical deployments, smart terminals collect sensor data across wide geographical areas (e.g., energy grids, agriculture, environmental monitoring) and periodically submit aggregated readings to a blockchain for settlement and automation. A particularly compelling use case emerges in healthcare data management, where hospitals across different regions need to securely submit patient statistics, research data, or epidemic monitoring information to a blockchain-based health information system. Due to stringent information security requirements and privacy regulations, each regional healthcare network typically designates only one trusted gateway node to communicate with the external blockchain network, while maintaining internal data aggregation among local hospitals within the region. This creates a natural two-layer architecture where intra-regional data collection and inter-regional consensus must be carefully coordinated.

While this architecture unlocks new cyber-physical and healthcare applications, it stresses traditional Byzantine Fault Tolerant (BFT) consensus in two ways: wide-area latency heterogeneity and high message concurrency. Flat BFT protocols such as PBFT and its derivatives exhibit quadratic communication in the number of participants and experience congestion-induced queuing delays under load, impairing both throughput and tail latency in WAN settings [1,2,3,4].

To address these challenges, we propose 2-layer EaZy BFT (2EZBFT), a hierarchical consensus architecture tailored for decentralized oracle networks with geographically distant smart terminals. 2EZBFT partitions nodes into small groups that first complete fast intra-group synchronization; group leaders then participate in an inter-group, linear voting procedure to finalize a global result with aggregated signatures. This two-phase design preserves BFT safety and liveness while reducing message fan-out and confining most traffic to low-latency local regions, similar in spirit to committee-based or multi-level approaches explored in recent consensus research [5,6,7,8].

Our contributions are threefold:(1)We formalize a geo-aware, two-layer BFT model for oracle networks and provide safety/liveness arguments under partial synchrony [9,10].(2)We design practical mechanisms for leader selection and evidence aggregation, leveraging verifiable randomness and compact BLS signatures to reduce inter-group costs [11,12].(3)We implement a discrete-event simulator and present a comparative evaluation against representative flat BFT protocols.

## 2. Related Works

Beyond consensus design, our study is motivated by the specific needs of oracle systems. Prior works including Town Crier and Chainlink established authenticated data feeds and decentralized oracle networks, while recent systems investigate privacy-preserving attestations and robust committee formation [5,13,14,15]. In the last few years, production oracle stacks have evolved toward off-chain aggregation and cross-chain delivery (e.g., OCR and CCIP), and new oracle networks (e.g., Pyth, UMA, API3) have broadened the design space [16,17,18]. Meanwhile, 6G-assisted security management and cloud–edge coordination provide complementary perspectives on securing wide-area cyber-physical systems [19]. In SDN-based Industrial IoT (IIoT) deployments, hybrid entropy and blockchain-based DDoS defense further illustrates the synergy between programmable networks and blockchain coordination [20]. 2EZBFT builds on these foundations with a communication-efficient, geo-aware BFT substrate that keeps the oracle pipeline high-performing and resilient across regions.

Decentralized oracle networks fetch, attest, aggregate, and deliver external data to smart contracts [13,14]. A typical pipeline includes the following: (i) request assignment (policy/reputation/stake/randomness); (ii) data collection with per-node attestations; (iii) off-chain aggregation (median/majority and optional BLS/threshold compression) [6,12]; and (iv) on-chain verification/consumption (e.g., OCR-style reporting and cross-chain delivery via CCIP) [14,16]. Production stacks (e.g., Pyth, UMA, API3) broaden the design space [17,18]. Geo-distributed deployments (e.g., one gateway per region) exacerbate WAN heterogeneity and congestion, stressing flat BFT coordination. Contemporary networking and system studies—such as slotted/wireless scheduling and SDN-based IIoT defenses—motivate bandwidth-efficient, geo-aware coordination [19,20,21,22]. In parallel, federated settings highlight the benefits of local computation with global aggregation, echoing our intra-/inter-group separation [23].

PBFT inaugurated practical BFT SMR with a three-phase protocol [1]. Zyzzyva introduced speculative execution to cut latency in fault-free executions [2]. HotStuff unified safety via chained commits with linear communication per view [3]. RBFT reduced leader bottlenecks through redundant instances, and SBFT combined signatures via collector roles [6]. Engineering libraries such as BFT-SMaRt accelerated adoption [4]. For empirical focus, we compare against PBFT (quadratic), Zyzzyva (optimistic), and HotStuff (linear) because they span the principal flat communication regimes most relevant to WAN queuing. RBFT/SBFT are valuable optimizations but do not fundamentally change the flat, cross-region broadcast pattern that inflates fan-out under WAN constraints.

Scalability is pursued via several orthogonal levers. Algorand pioneered randomized committees and VRF-based leader election [5]. Tendermint/CometBFT provides practical instant finality for app-specific chains [24,25]. HoneyBadgerBFT achieves asynchrony with cryptographic primitives, trading latency for partition tolerance [26]. SBFT lowers steady-state costs using collectors and threshold signatures [6]. Mir-BFT increases throughput by parallelizing agreement across instances [7]. DAG-backed designs (Narwhal/Tusk, Bullshark) decouple mempool dissemination from consensus [8,27]. While these lines advance scalability (committee sampling, asynchrony, parallel instances, DAG-ordering), our target is distinct: WAN geo-heterogeneity in oracle deployments where message fan-out and inter-region congestion dominate. Hence, our baselines emphasize flat PBFT/Zyzzyva/HotStuff to isolate the effect of hierarchy on queuing under identical WAN models.

Compact aggregation with BLS amortizes verification and reduces bandwidth [12]; VRFs provide publicly verifiable leader selection without coordination [11]. Committee sampling and threshold signatures are standard tools for lowering message costs in Byzantine settings [5,6]. 2EZBFT employs BLS-style aggregation for intra-/inter-group commitments and verifiable randomness for unbiased, replay-resistant leader rotation.

Town Crier explored authenticated feeds from HTTPS using trusted hardware [13]. Chainlink popularized decentralized oracle networks with on-chain/off-chain components and reputation-driven node selection [14]. Additional systems studied privacy-preserving attestations and scalable committees for oracles and payments [15,28]. In contrast, 2EZBFT focuses on the consensus substrate for geo-distributed oracles: it explicitly models WAN heterogeneity and queuing costs of flat message patterns and shows that a two-layer hierarchy sustains latency/throughput under load while preserving BFT safety/liveness.

## 3. Two-Layer EaZy BFT Overview

This section details our two-phase hierarchical consensus algorithm, designed for efficiency and Byzantine fault tolerance in large-scale decentralized oracle networks.

### 3.1. Design of 2-Layer EaZy BFT

We consider a network of *N* nodes partitioned into *g* groups, $G=\{G_{1},G_{2},\ldots ,G_{g}\}$. Each group $G_{i}$ contains $k_{i}$ nodes, $N_{i}=\left\{n_{i1},n_{i2},\ldots ,n_{ik_{i}}\right\}$, and can tolerate up to $f_{i}$ Byzantine nodes, where $k_{i}\geq 2f_{i}+1$. Each group has a leader $L_{i}$, and a primary leader $L_{p}$ coordinates inter-group consensus. The system operates in rounds *R*, with hierarchical view numbers *V* (global) and $V_{i}$ (group-specific) to manage leader changes. Together, these variables ensure both freshness against replay attacks (via *R*) and liveness through fault recovery (via *V* and $V_{i}$). The symbols and variables used in the paper are listed in Table 1.

For blockchain oracles, data collection tasks may be initiated by smart contracts or timed tasks. In 2EZBFT, the consensus process unfolds in two main phases.

**Phase 1: Intra-group Synchronization (Algorithm 1):** Each group independently generates a consensus value $C_{i}$. This phase includes a cross-verification commitment step to prevent double-value attacks.**Phase 2: Inter-group Linear Consensus (Algorithm 2):** All group leaders who have completed intra-group synchronization submit their results to the primary leader $L_{p}$, who aggregates and verifies all group consensus values. All valid group leaders then participate in a linear voting and signature aggregation process to finalize the global consensus.

As listed in Table 1, $L_{i}$ denotes the group leader of group $G_{i}$, while $n_{ij}$ denotes node *j* in group $G_{i}$. The architectural design of 2EZBFT is shown in Figure 1 and the workflow of 2EZBFT is shown in Figure 2.

**Design Rationale for Intra-group Synchronization:** Unlike traditional Byzantine consensus protocols that require multiple voting rounds, our Phase 1 algorithm uses a *leader-aggregation* approach where the group leader directly computes consensus after collecting sufficient data. This design is justified because of the following:**Oracle Data Characteristics:** Oracle data from external sources (e.g., price feeds, weather data) typically have objective truth, making simple aggregation (median/majority) sufficient for conflict resolution**Cryptographic Proof:** The aggregated commitment on $C_{i}$ contains signatures from a majority of nodes, providing non-repudiation without requiring interactive voting**Linear Complexity:** Eliminates the $O(k_{i}^{2})$ message complexity of traditional PBFT within groups
**Algorithm 1:** Phase 1: intra-group synchronization.
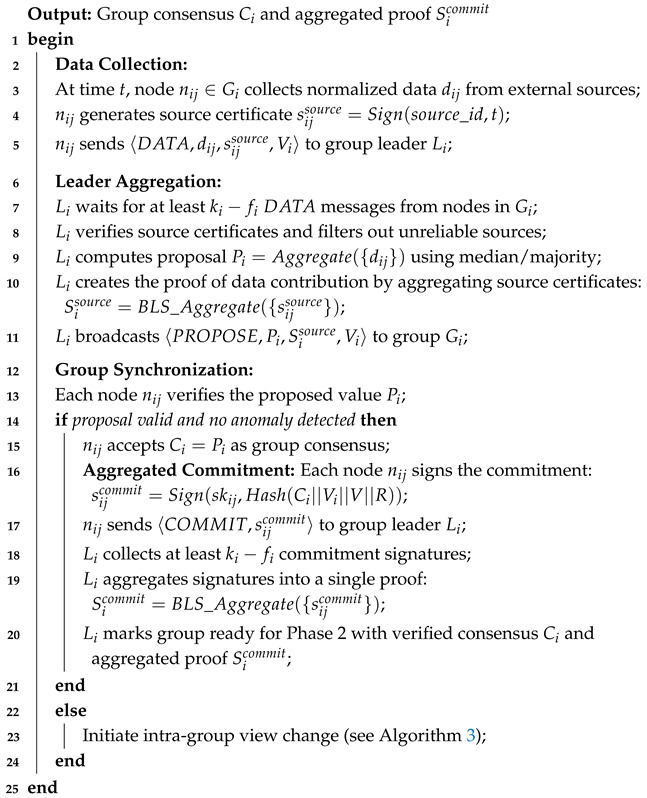


In this phase, node $n_{ij}$ collects data $d_{ij}$ from external sources and generates a source certificate $s_{ij}^{source}$, which is sent to the group leader $L_{i}$. The group leader aggregates the data from at least $k_{i}-f_{i}$ nodes in the group, computes a proposal value $P_{i}$, and broadcasts to nodes in group $G_{i}$. Node $n_{ij}$ verifies whether $P_{i}$ is computed from at least $k_{i}-f_{i}$DATA messages through $S_{i}^{source}$. When $P_{i}$ is accepted by nodes in group $G_{i}$, each node signs the commitment message with its private key, creating a commitment signature $s_{ij}^{commit}$. The group leader $L_{i}$ collects these signatures and aggregates them into a single proof of commitment $S_{i}^{commit}$. This aggregated proof is then used in Phase 2 to ensure that the group consensus is valid and prevents double-value attacks.
**Algorithm 2:** Phase 2: inter-group consensus (normal operation).
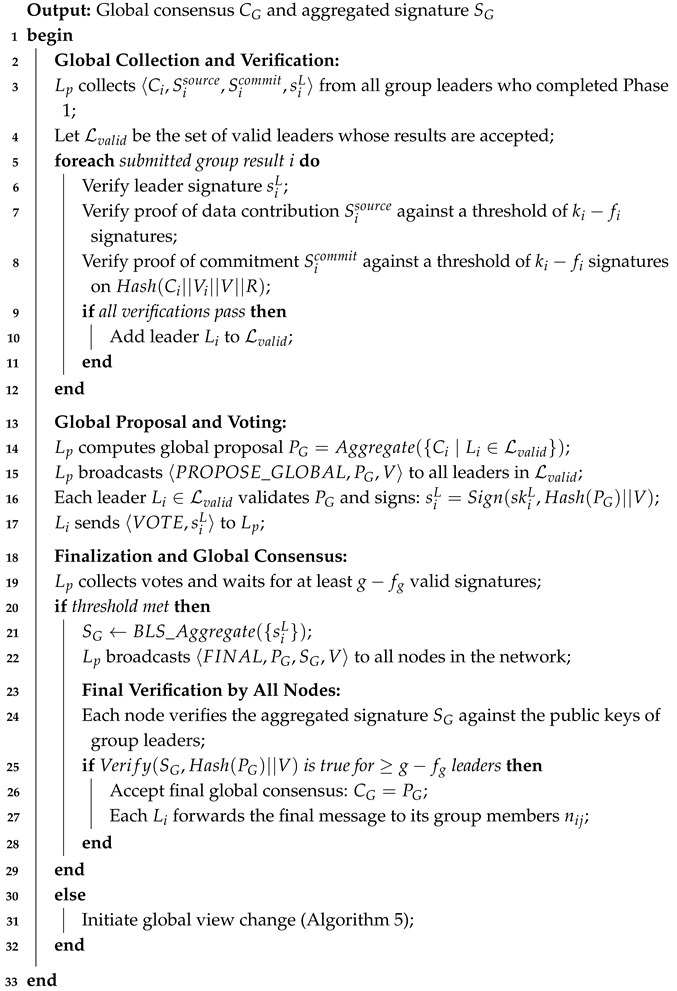


In this phase, the primary leader $L_{p}$ collects all group results from valid leaders and verifies them. It then computes a global proposal $P_{G}$ by aggregating the group consensus values. The primary leader broadcasts this proposal to all valid leaders. Each leader $L_{i}$ validates $P_{G}$ and signs a leader signature $s_{i}^{L}$. $L_{p}$ collects the signatures and waits for at least $g-f_{g}$ valid signatures. If the threshold is met, it aggregates the signatures into a single proof $S_{G}$ and broadcasts the final consensus message to all nodes. Each node verifies the aggregated signature against the public keys of group leaders. If the verification passes, the final global consensus $C_{G}$ is accepted. The smart contracts on blockchain can verify the value with the aggregated signatures then use this consensus value for further processing, such as triggering events or executing transactions.

Algorithm 3 describes the process for handling a Byzantine group leader. If any node detects an anomaly, such as an invalid proposal or a timeout, it broadcasts a challenge message within its group. To prevent malicious nodes from disrupting the network, a view change is triggered only when at least $f_{i}+1$ nodes issue a challenge. This threshold ensures that at least one honest node concurs with the accusation. The new leader is deterministically and verifiably selected using the global signature from the previous round ($S_{G}^{R-1}$), the new group view number ($V_{i}$), and the round number (*R*) as a seed. This prevents Byzantine nodes from influencing the leader selection process and ensures a swift and orderly recovery within the group.
**Algorithm 3:** Intra-group view change (Byzantine group leader detection).
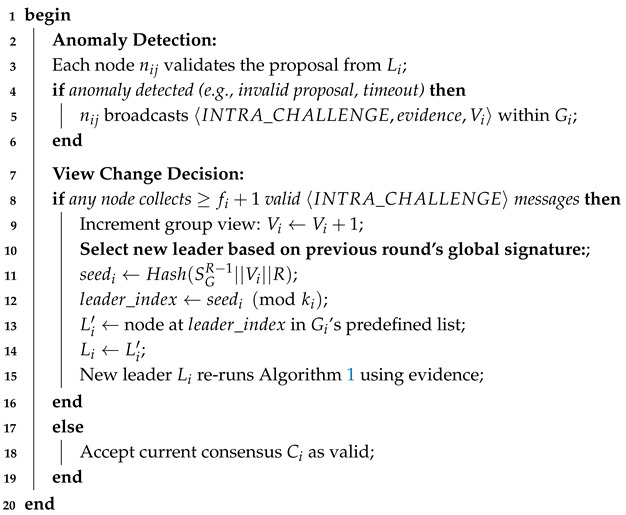


Similarly, Algorithm 4 outlines the procedure for replacing a faulty primary leader. If the primary leader fails to generate a valid global proposal or experiences significant delays, the valid group leaders broadcast a global challenge. A global view change is initiated upon collecting $f_{g}+1$ such challenges, ensuring the decision is supported by at least one honest group leader. A new primary leader is then selected from the pool of group leaders using a deterministic mechanism based on the previous round’s global signature. This hierarchical approach ensures that faults are handled at the appropriate level, minimizing disruption to the overall network. For a unified summary of the two-level recovery workflow that integrates both intra-group and global procedures, see Algorithm 5.
**Algorithm 4:** Global view change (Byzantine primary leader detection).
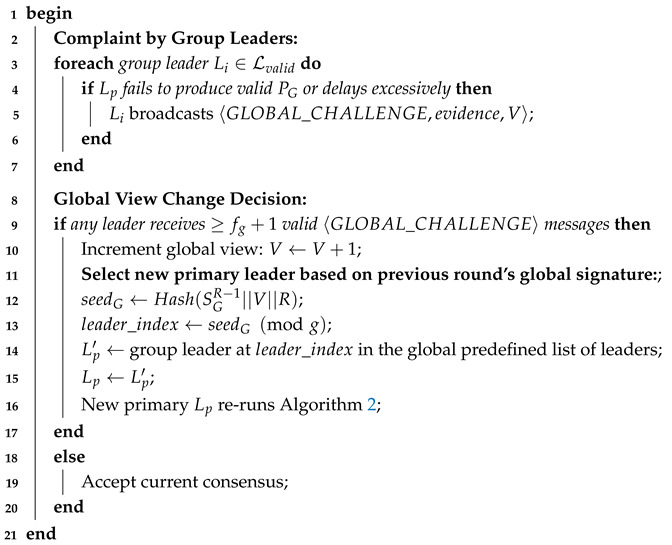


Algorithm 5 unifies recovery at two layers with clear triggers and deterministic leader reselection. At the group layer, any node that detects an anomaly (timeout or invalid proposal) can issue an INTRA_CHALLENGE; when a node gathers at least $f_{i}+1$ valid challenges for the same $\left(V_{i},R\right)$, the group increments $V_{i}$ and deterministically elects a new leader using a seed derived from the previous round’s global signature $S_{G}^{R-1}$, the group view $V_{i}$, and the round *R*. Crucially, this leaves the global view *V* unchanged, confining the repair to the affected group. At the global layer, leader anomalies similarly trigger GLOBAL_CHALLENGE messages; upon collecting at least $f_{g}+1$ valid challenges for the same (V,R), the system increments *V* and deterministically rotates the primary leader among group leaders using the same seed construction, while preserving all groups’ local views $V_{i}$. The $f_{i}+1$ and $f_{g}+1$ thresholds ensure that at least one honest party endorses a view change, preventing spurious reconfigurations; the decoupled views ($V_{i}$ vs. *V*) bound fault impact and expedite progress.
**Algorithm 5:** Hierarchical view management and recovery protocol.
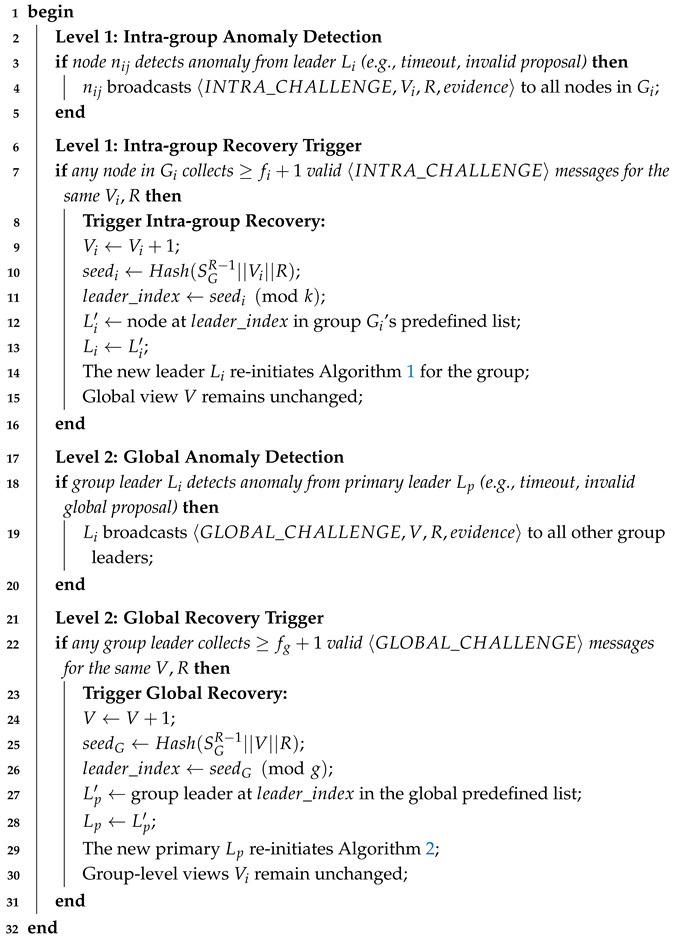


### 3.2. Security and Performance Analysis

This section provides a detailed analysis of our proposed 2-layer EaZy BFT (2EZBFT) consensus algorithm in terms of security and performance, comparing it with leading BFT algorithms. We also discuss implications for data availability in storage-constrained settings, where erasure-coded blockchain storage and transport-level optimizations such as trigger-based ARQ can complement hierarchical consensus [29].

#### 3.2.1. Security Analysis

2EZBFT ensures robust security through its hierarchical architecture and carefully designed mechanisms. We prove its core consensus properties—safety and liveness—formally.

##### Safety (Agreement)

Safety guarantees that all honest nodes agree on the same value.

**Lemma** **1**(**Intra-group Safety**)**.** *For any honest group $G_{i}$, all honest nodes $n_{ij}$ within that group will agree on the same group consensus $C_{i}$ for a given round R and view $V_{i}$.*

**Proof.** A group leader $L_{i}$ requires $k_{i}-f_{i}$ commitment signatures ($s_{ij}^{commit}$) to form the aggregated proof $S_{i}^{commit}$. Given the assumption $k_{i}\geq 2f_{i}+1$, any two quorums of size $k_{i}-f_{i}$ must have at least one honest node in their intersection. Since an honest node will only sign one proposal value $P_{i}$ for a given $\left(R,V_{i}\right)$, it is impossible for an honest leader to create two different valid proofs $S_{i}^{commit}$ for two different proposals. A Byzantine leader cannot forge the signatures of $k_{i}-f_{i}$ nodes. Therefore, any valid $S_{i}^{commit}$ uniquely corresponds to a single consensus value $C_{i}$, ensuring agreement within the group. □

**Theorem** **1**(**Global Safety**)**.** *For any given round R and global view V, all honest nodes in the network will agree on the same final global consensus $C_{G}$.*

**Proof.** This property follows from Lemma 1. The primary leader $L_{p}$ must collect $g-f_{g}$ valid group consensus packages $\left(\langle C_{i},S_{i}^{commit}\rangle \right)$. By Lemma 1, each valid $S_{i}^{commit}$ from an honest group is unique and unforgeable. The final global signature $S_{G}$ requires signatures from $g-f_{g}$ group leaders. Given $g\geq 2f_{g}+1$, any two global voting quorums of size $g-f_{g}$ also have an intersection of at least one honest leader. An honest leader will only vote for one global proposal $P_{G}$ for a given (R,V). Thus, it is impossible to generate two different valid global signatures $S_{G}$. This guarantees that the final consensus $\left(C_{G},S_{G}\right)$ is unique across the entire network. □

##### Liveness (Termination)

Liveness guarantees that all honest nodes will eventually decide on a value.

**Lemma** **2**(**Intra-group Liveness**)**.** *If the leader $L_{i}$ of a group $G_{i}$ is Byzantine, the honest nodes in $G_{i}$ will eventually elect a new, honest leader in a finite number of view changes.*

**Proof.** If a leader $L_{i}$ is faulty (e.g., fails to send proposals or sends invalid ones), honest nodes $n_{ij}$ will not receive a valid proposal and will eventually time out. As per Algorithm 5, they will broadcast an 〈INTRA_CHALLENGE〉. Once $f_{i}+1$ such challenges are collected, a view change is triggered, and the group view $V_{i}$ is incremented. The new leader is selected deterministically based on $Hash(S_{G}^{R-1}||V_{i}||R)$. Since there are a finite number of nodes ($k_{i}$) in the group and at least one is honest, this process must eventually result in an honest node being selected as the leader, allowing the consensus process to proceed. □

**Theorem** **2**(**Global Liveness**)**.** *The network as a whole will eventually produce a final global consensus $C_{G}$ in a finite amount of time.*

**Proof.** This property follows from Lemma 2. If the primary leader $L_{p}$ is Byzantine, honest group leaders will detect this failure and broadcast 〈GLOBAL_CHALLENGE〉. A global view change is triggered after $f_{g}+1$ challenges. Similar to the intra-group case, a new, honest primary leader will eventually be elected. An honest $L_{p}$ will wait for group leaders to submit their consensus values. By Lemma 2, each group is guaranteed to eventually produce a consensus value. Once the honest $L_{p}$ receives $g-f_{g}$ group results, it will faithfully execute the rest of the protocol, ensuring that a final global consensus is eventually reached and broadcast to all nodes. □

##### Byzantine Fault Tolerance

The overall fault tolerance of 2EZBFT depends on the distribution of Byzantine nodes. We analyze the system’s tolerance threshold from a worst-case perspective, where an attacker concentrates resources to attack the weakest point.

**Theorem** **3**(**System Fault Tolerance Threshold**)**.**
*Let the system have N nodes, partitioned into g groups of size k each (i.e., N=gk). The total number of Byzantine nodes F that the system can tolerate is given by: $F<\min(f_{i}+1,f_{g}+1)$ where $f_{i}=\left\lfloor \left(k-1\right)/2\right\rfloor$ is the maximum number of Byzantine nodes tolerated within a single group, and $f_{g}=\left\lfloor \left(g-1\right)/2\right\rfloor$ is the maximum number of Byzantine nodes tolerated in the network of group leaders. In other words, the system remains secure as long as the total number of Byzantine nodes F is less than the number required to compromise any single group ($f_{i}+1$) and also less than the number required to compromise the inter-group network ($f_{g}+1$).*

**Proof.** To break the system’s consensus, an adversary has two primary strategies:
**Concentrated Attack Strategy:** The adversary concentrates all Byzantine nodes into a single group $G_{i}$ to corrupt its intra-group consensus. According to Lemma 1 (Intra-group Safety), compromising a group of size *k* requires controlling $f_{i}+1$ nodes, where $f_{i}=\left\lfloor \left(k-1\right)/2\right\rfloor$. If the adversary’s total number of nodes $F<f_{i}+1$, this attack cannot succeed.**Distributed Attack Strategy:** The adversary distributes Byzantine nodes across different groups and attempts to make them all group leaders to corrupt the global consensus. According to Theorem 1 (Global Safety), corrupting the global consensus requires controlling $f_{g}+1$ group leaders, where $f_{g}=\left\lfloor \left(g-1\right)/2\right\rfloor$. If the adversary’s total number of nodes $F<f_{g}+1$, they cannot break the global consensus even if all *F* nodes become group leaders.
Since an attacker can choose either strategy, the system’s security depends on the weakest link. Therefore, as long as the total number of Byzantine nodes *F* satisfies both $F<f_{i}+1$ and $F<f_{g}+1$, the system is guaranteed to be secure. □

**Corollary** **1**(**Optimal Grouping Strategy**). *For a fixed total number of nodes N, the system’s overall fault tolerance F is maximized when the group size k and the number of groups g are approximately equal (i.e., $k\approx g\approx \sqrt{N}$). This configuration balances the difficulty of concentrated and distributed attacks, making the values of $f_{i}+1$ and $f_{g}+1$ roughly equal, thereby increasing the overall Byzantine fault tolerance threshold of the system.*

#### 3.2.2. Complexity Analysis

We compare the complexity of 2EZBFT with PBFT, Zyzzyva, and HotStuff under normal operating conditions. Let *N* be the total number of nodes, *g* be the number of groups, and *k* be the size of each group (N=g×k). The complexity comparison results are shown in Table 2.

**Communication Rounds**: 2EZBFT requires 7 rounds of communication in the normal case (3 for intra-group and 4 for inter-group consensus). While this is higher than the 3 rounds typical of flat BFT protocols, these rounds involve significantly fewer messages and are structured to minimize network congestion, as analyzed below.**Message Complexity**: The total number of messages per consensus instance is O(N).-**Phase 1 (Intra-group)**: In each of the *g* groups, nodes send data to the leader (k−1 messages), the leader broadcasts a proposal (k−1 messages), and nodes send signatures back (k−1 messages). This totals g×3(k−1)≈3N messages.-**Phase 2 (Inter-group)**: Group leaders send proposals to the primary (g−1 messages), the primary broadcasts a global proposal (g−1 messages), leaders send votes back (g−1 messages), and the primary broadcasts the final result (g−1 messages). This is approximately 4g messages.-**Total**: The complexity is O(3N+4g+N). Since g≪N, the overall message complexity is dominated by the intra-group communication, resulting in O(N).

#### 3.2.3. Formal Latency Analysis

To formalize the latency advantage in realistic, congestion-prone networks, we model the effective delay of a communication round, $T_{\mathrm{round}}$, as the sum of propagation delay ($T_{\mathrm{prop}}$), processing delay ($T_{\mathrm{proc}}$), and queuing delay ($T_{\mathrm{queue}}$).$$T_{\mathrm{round}}=T_{\mathrm{prop}}+T_{\mathrm{proc}}+T_{\mathrm{queue}}$$We assume $T_{\mathrm{prop}}=\Delta$ (a constant) and $T_{\mathrm{proc}}$ is negligible for simplicity. The critical factor is $T_{\mathrm{queue}}$, which grows non-linearly with the number of concurrent messages, *M*. Based on queuing theory, which shows that latency grows explosively as traffic approaches network capacity, we can approximate this with a super-linear model, such as a quadratic function: $T_{\mathrm{queue}}\left(M\right)\approx \gamma M^{2}$, where γ is a coefficient representing network sensitivity to congestion.

For a flat BFT algorithm with 3 rounds, the total number of concurrent messages in each round is $M_{\mathrm{flat}}\approx N$. The total latency, $L_{\mathrm{flat}}$, is dominated by the queuing delay:$$L_{\mathrm{flat}}=3\cdot T_{round\_flat}\approx 3\left(\Delta +\gamma N^{2}\right)=3\Delta +3\gamma N^{2}$$

For 2EZBFT, we have two phases. In Phase 1 (3 rounds), communication is confined within groups of size *k*, so $M_{\mathrm{intra}}\approx k$. In Phase 2 (4 rounds), communication is among *g* leaders, so $M_{\mathrm{inter}}\approx g$. The total latency, $L_{2\mathrm{EZBFT}}$, is:$$\begin{array}{cc} L_{2\mathrm{EZBFT}} & =3\cdot T_{round\_intra}+4\cdot T_{round\_inter}\\ \; & \approx 3\left(\Delta +\gamma k^{2}\right)+4\left(\Delta +\gamma g^{2}\right)\\ \; & =7\Delta +\gamma (3k^{2}+4g^{2}) \end{array}$$

To find when 2EZBFT is faster, we compare the dominant latency terms:$$\gamma \left(3k^{2}+4g^{2}\right)<3\gamma N^{2}$$Assuming an optimal distribution where $k\approx g\approx \sqrt{N}$, we substitute this into the inequality:$$3{\left(\sqrt{N}\right)}^{2}+4{\left(\sqrt{N}\right)}^{2}<3N^{2}$$$$7N<3N^{2}$$This inequality holds for any N>7/3. This formal analysis demonstrates that as the network size *N* grows, the quadratic term $3\gamma N^{2}$ in flat architectures quickly dwarfs the linear term γ(7N) in our hierarchical architecture. Therefore, by mitigating network congestion, 2EZBFT’s architectural advantage in realistic networks outweighs its higher number of communication rounds.

## 4. Experimental Evaluation

To validate the performance of 2EZBFT, we designed a series of experiments to compare it against leading BFT algorithms in a simulated, realistic network environment. Our evaluation focused on key performance indicators such as throughput and latency under various network conditions and workloads. Unless otherwise stated, we distinguish two evaluation modes and report which one a figure uses: (i) strong scaling (fixed offered load while varying the number of replicas), and (ii) weak scaling (offered load grows proportionally with system size to keep per-replica pressure approximately constant).

### 4.1. Simulation Environment

All experiments were conducted within a custom-built, discrete-event simulator written in Go, allowing for precise control over network conditions and node behavior.

#### 4.1.1. Benchmark Algorithms

To provide a focused comparison that highlights the core architectural benefits of 2EZBFT, we selected three representative BFT algorithms that embody distinct communication paradigms:**PBFT**: The foundational BFT consensus algorithm with quadratic message complexity ($O(N^{2})$), representing traditional flat BFT approaches and serving as a baseline for performance comparison.**Zyzzyva**: An optimistic BFT algorithm that achieves linear complexity (O(N)) under normal conditions but reverts to PBFT-like behavior under faults, allowing us to evaluate both optimistic and pessimistic execution paths.**HotStuff**: A modern, leader-based BFT protocol with linear communication complexity and streamlined view changes, representing the current flat BFT design.

These three protocols were selected to span the principal *flat* communication regimes—quadratic (PBFT), optimistic (Zyzzyva), and linear (HotStuff)—so that improvements could be attributed to hierarchy rather than to orthogonal features. Protocols centered on stake-based committee sampling (e.g., Algorand), strong asynchrony (HoneyBadgerBFT), or multi-instance parallelization (Mir) optimize for different axes and would confound a WAN-focused baseline comparison. Practical round-based PoS BFT (Tendermint/CometBFT) follows the same cross-region broadcast pattern as HotStuff in our model. Collector/redundancy variants (SBFT/RBFT) retain flat coordination and are partly subsumed by the linear baseline.

#### 4.1.2. Hardware and Software

To ensure the reproducibility and relevance of our findings on commonly available hardware, the simulation was executed on a desktop machine equipped with an Intel Core i5-9600K processor (six cores, six threads), 32 GB of DDR4 RAM, and a 2 TB NVMe SSD, running Ubuntu 22.04 LTS. This setup represents a typical mid-range consumer-grade machine, allowing us to evaluate the algorithms’ performance in a practical, non-idealized environment.

#### 4.1.3. Network Model

To assess performance in a realistic WAN setting, we simulated a geographically distributed deployment across four distinct regions (Regions A, B, C, D). The network latency between these regions was modeled using a standard, asymmetric latency matrix, as shown in Table 3, with an additional random jitter of ±5 ms per message. The latency within the same region was set to 5 ms. All network links were configured with a bandwidth of 100 Mbps and a random packet loss rate of 0.1% to mimic real-world internet conditions.

#### 4.1.4. Node Configuration and Scalability

We evaluated the algorithms across different network sizes, with the total number of nodes *N* set to 16, 32, 64, and 128. For 2EZBFT, we adopted a geo-aware grouping strategy where nodes within the same geographical region were clustered into the same group. For instance, in a 64-node setup, we used a configuration of g=4 groups with k=16 nodes each, with each group corresponding to one geographic region. This setup is designed to leverage 2EZBFT’s hierarchical structure by localizing high-frequency intra-group communication within low-latency regional networks.

### 4.2. Simulation Implementation

To enable comprehensive performance comparison against established BFT protocols (PBFT, Zyzzyva, HotStuff), we developed a discrete-event simulator in Go that faithfully implemented all compared algorithms under identical network conditions.

#### 4.2.1. Simulator Architecture

The simulator employed a centralized event scheduler using a priority queue to maintain strict chronological ordering of all network events. Each algorithm was implemented as a separate state machine, allowing direct performance comparison under identical network conditions. The simulator modeled both intra-region (5 ms latency) and inter-region communication using the asymmetric latency matrix shown in Table 3, with additional jitter (±5 ms) to simulate realistic WAN conditions.

#### 4.2.2. Workload Model and Scaling Modes

Our oracle workload comprised fixed-size transactions (payload 256 B unless otherwise noted) with constant-time cryptographic placeholders and a fixed batch size of 64 per proposal. Unless otherwise stated, intra-group inputs $d_{ij}$ are understood as normalized readings of the same external variable observed by honest nodes within a short decision window, so that simple aggregation (e.g., median/majority) is meaningful; this mirrors the common oracle setting of price or sensor feeds and aligns with the design assumption used in Section 2. We adopted two standard scaling modes: (1) *strong scaling* kept the total offered load independent of *N* to reveal the intrinsic scalability of single-committee protocols; (2) *weak scaling* increased the offered load in proportion to the system size/parallelism. In our hierarchical 2EZBFT, parallelism corresponds to the number of groups/committees participating concurrently. For weak scaling, the number of clients (or request generation rate) was set so that the per replica offered load remained approximately constant as *N* grew.

#### 4.2.3. Protocol Implementation

Each BFT algorithm maintained its authentic message flow:**PBFT**: Implements the complete three-phase protocol (pre-prepare, prepare, commit) with quadratic message complexity;**Zyzzyva**: Models optimistic execution with fallback to PBFT under faults;**HotStuff**: Implements the linear three-phase chain (prepare, pre-commit, commit, decide);**2EZBFT**: Executes the two-layer protocol as described in Algorithms 1 and 2.

#### 4.2.4. Fault Injection and Load Testing

The simulator supported controlled Byzantine fault injection at three levels: (i) follower node faults within groups; (ii) group leader failures triggering intra-group view changes; (iii) primary leader failures requiring global view changes. Load testing was implemented through configurable request rates, enabling evaluation under both low-load (single request per round) and high-load conditions. Geographical deployment strategies were modeled by varying group-to-region mapping, allowing evaluation of 2EZBFT’s geo-aware optimizations.

#### 4.2.5. Metrics Collection and Validation

All events were logged with precise timestamps enabling measurement of end-to-end consensus latency (from first data collection to final decision delivery) and system throughput (completed consensus rounds per second). The simulator abstracted cryptographic operations using constant-time placeholders while preserving all protocol dependencies and quorum requirements. This approach isolated communication-pattern effects from computational overhead, enabling fair comparison of protocol structural efficiency.

### 4.3. Simulation Results

Our evaluation focused on communication patterns and queuing behavior under WAN heterogeneity using a discrete-event simulator with constant-time cryptographic placeholders and fixed-size payloads. The intra-group aggregation in the experiments implicitly presumed that honest nodes observed the same external variable and that their inputs were normalized over a short decision window; when this condition is violated (e.g., heterogeneous data sources, stale or region-specific views), simple aggregators can be biased unless complemented by application-layer defenses such as source diversity, outlier filtering, or stake and reputation weighting. The simulator abstracted transport intricacies and hardware parallelism and did not model adversarial cryptographic costs or DoS at the networking layer; therefore, absolute numbers should be interpreted as directional and comparative rather than as deployment-ready benchmarks. A broader head-to-head against additional protocols under harmonized assumptions, and validation on real WAN testbeds with geo-aware placement, are left to future work.

#### 4.3.1. Performance vs. Network Size

We first evaluated how the algorithms scaled as the number of nodes in the network increased from 16 to 128.

**Throughput (Figure 3):** As shown in Figure 3, the throughput of flat architectures like PBFT, Zyzzyva, and HotStuff degrades as the network size grows due to increased communication overhead. In contrast, 2EZBFT’s throughput scales positively, as its hierarchical structure effectively partitions the network and allows for parallel processing within groups. This demonstrates its suitability for large-scale deployments.**Latency (Figure 4 and Figure 5):** We measured latency under both low- and high-load conditions (using closed-loop clients). Under *low load*, 2EZBFT exhibits a modestly higher base latency because it performs seven communication rounds (three intra-group + four inter-group), whereas flat BFTs completes in three rounds. Under *high load*, however, flat protocols accumulate substantial queuing delay at the primary (and along all-to-all broadcasts), as the leader’s processing/broadcast capacity saturates; their latency grows super-linearly with *N*. In contrast, 2EZBFT partitions the offered load across groups so that most messages queue within small committees and only aggregated results traverse the inter-group layer, keeping queuing delay bounded and the latency curve flatter as *N* increased.

##### Why Do Low-Load and High-Load Latency Behave Differently?

The contrasting latency behaviors arise from two distinct dominant factors:**Round count dominates at low load.** When the system is far from saturation, propagation delay per round (≈Δ) dominates and queuing is negligible. 2EZBFT uses seven rounds (Section 4), so its base latency is roughly $\frac{7}{3}$ that of a three-round flat BFT in the same network.**Queuing dominates at high load.** As the offered load approaches component capacity, queuing delay $T_{\mathrm{queue}}$ grows super-linearly with concurrent messages *M* (Section 3.2.3). In flat BFTs, one leader and all-to-all broadcasts create large M∼N, causing sharp latency inflation once the primary becomes the bottleneck. 2EZBFT confines heavy traffic to groups of size *k* and sends only aggregated artifacts across *g* leaders, so the heavy-traffic terms scale with *k* and *g* rather than *N*; thus the queuing component remains much smaller even though 2EZBFT has more rounds.

This explains why 2EZBFT may be slower under light traffic (more rounds) but becomes faster and more stable under high load and larger *N* (less queuing), matching the trends in Figure 4 and Figure 5.

#### 4.3.2. Impact of Byzantine Faults on 2EZBFT

To assess the robustness of 2EZBFT, we simulated three distinct Byzantine fault scenarios for a 64-node network.

**Follower Faults (Figure 6):** We introduced an increasing number of Byzantine follower nodes within each group (g=4, k=16). As shown in Figure 6, latency increases linearly but moderately, as faults are contained within groups and do not disrupt the global consensus process.**Group Leader Faults (Figure 7):** We simulated the failure of group leaders in a network with g=16 and k=4. Figure 7 shows that latency increases more significantly with each faulty leader, as this triggers intra-group view changes. However, the system remains operational and recovers, demonstrating the effectiveness of the view change protocol.**Primary Leader Fault (Figure 8):** The most severe scenario involves the failure of the primary leader. Figure 8 illustrates the latency impact of a primary leader failure compared to normal operation across various group structures. A failure incurs a substantial but fixed-time penalty for the global view change, after which the system swiftly elects a new primary leader and resumes operation.

#### 4.3.3. Impact of Group Structure on 2EZBFT

Finally, we investigated how the choice of group structure (*g* and *k*) affected performance for a fixed network size of N=64. As shown in Figure 9, there was a clear trade-off. Structures with fewer, larger groups (e.g., g=4, k=16) had higher latency due to increased intra-group communication complexity. Conversely, structures with more, smaller groups (e.g., g=16, k=4) had higher latency due to the overhead of inter-group coordination. The experiment confirmed our theoretical analysis, indicating that an optimal, balanced structure (e.g., g=8, k=8) exists that minimizes latency by balancing these two factors.

## 5. Conclusions

We presented 2-layer EaZy BFT (2EZBFT), a hierarchical Byzantine fault-tolerant consensus tailored for decentralized oracle networks with geographically distant smart terminals. By partitioning nodes into small groups for fast intra-group synchronization and performing linear inter-group voting with aggregated evidence, 2EZBFT reduces message fan-out and alleviates WAN congestion while preserving safety and liveness.

Analysis showed linear message complexity with hierarchical rounds (three intra + four inter) and a queuing-delay model where congestion grew with a fan-out, favoring hierarchy beyond modest network sizes. Simulations over an asymmetric WAN latency matrix indicated up to 45% higher throughput than flat baselines under load, with robustness to follower faults, group leader failures, and primary leader view changes. Sensitivity to group structure revealed a balanced configuration (e.g., $g\approx k\approx \sqrt{N}$) minimized latency, aligning with our theoretical insight.

In summary, 2EZBFT achieved up to 45% higher throughput under load than flat baselines, with aggregate capacity increasing under weak scaling via parallel groups. Although its base latency was slightly higher at low load (seven vs. three rounds), latency grew markedly more slowly under high load and larger *N*, consistent with the queuing analysis and simulator results. The protocol maintained operation under follower faults, group-leader failures, and primary-leader view changes with bounded recovery time. It retained linear message complexity of O(N) with hierarchical (3+4) rounds, reducing WAN fan-out relative to flat broadcasts. Performance was sensitive to group structure, and a balanced configuration (e.g., $g\approx k\approx \sqrt{N}$) minimized latency, matching both analysis and measurements.

Intra-group aggregation assumes honest nodes observe and normalize the same external variable within a short decision window; strong divergence can bias simple aggregators and should be mitigated at the application layer (e.g., source diversity, outlier filters, stake/reputation). At very low load, extra rounds dominate, reducing any latency advantage over flat three-round protocols. Benefits depend on balanced groups and geo-aware mapping; severe imbalance or poor mapping can erode gains. Cryptographic aggregation reduces bandwidth and on-chain size, but verification still incurs compute overhead. Finally, our evaluation was conducted in a simulator with constant-time cryptographic placeholders and a WAN latency model; broader head-to-head comparisons with SBFT/RBFT/Mir/Algorand/HoneyBadger/Tendermint and closely related hierarchical designs remain future work.

To address the above constraints and strengthen deployability, we plan to (i) deploy a geo-aware prototype on real testbeds with controlled WAN latencies to validate the queuing model and sensitivity to geographic mapping; (ii) integrate DAG-backed dissemination to decouple data availability from agreement and reduce tail latency under bursty traffic; (iii) design adaptive grouping and leader rotation that track regional network/quality shifts to preserve balance and resilience; (iv) incorporate economic incentives and Sybil resistance into leader and group selection to harden against strategic adversaries and churn; and (v) formally verify state machines and recovery logic to eliminate corner-case safety and liveness issues. We believe 2EZBFT provides a practical foundation for reliable, efficient data interaction between blockchains and the physical world.

## Figures and Tables

**Figure 1 sensors-25-06268-f001:**
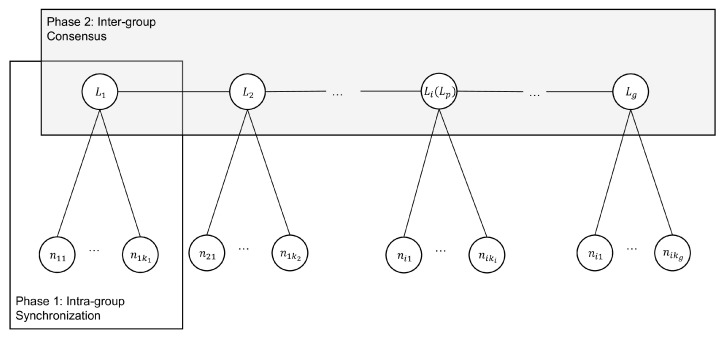
2EZBFT architectural diagram.

**Figure 2 sensors-25-06268-f002:**
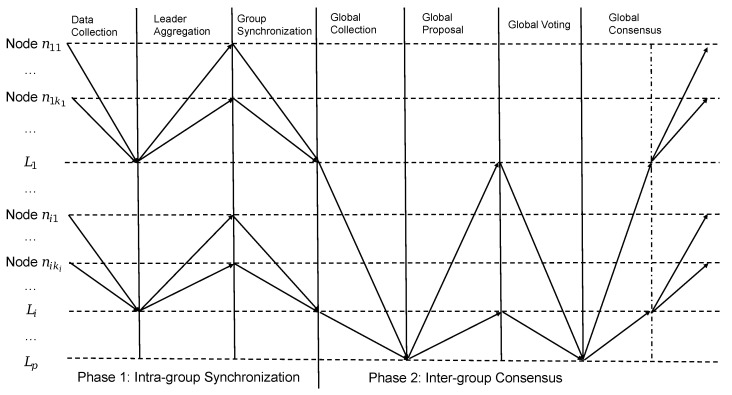
2EZBFT workflow overview.

**Figure 3 sensors-25-06268-f003:**
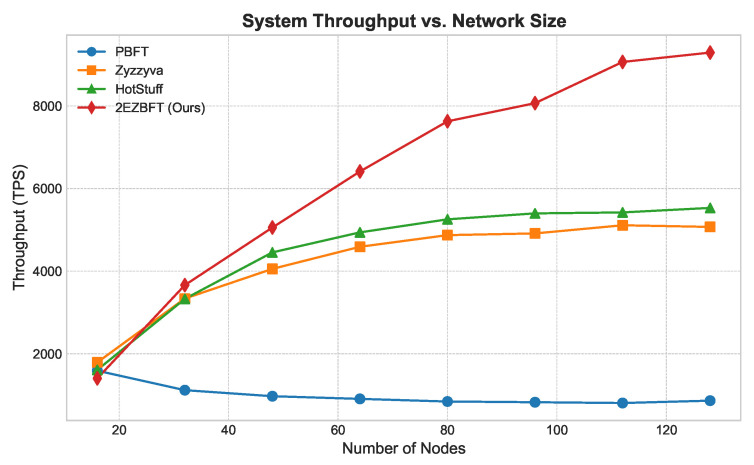
System throughput vs. network size (weak scaling). Total offered load grows proportionally with system size/parallelism. 2EZBFT leverages hierarchical groups to execute in parallel, increasing aggregate capacity; flat single-committee baselines (PBFT, Zyzzyva, HotStuff) are shown in their standard single-committee configurations.

**Figure 4 sensors-25-06268-f004:**
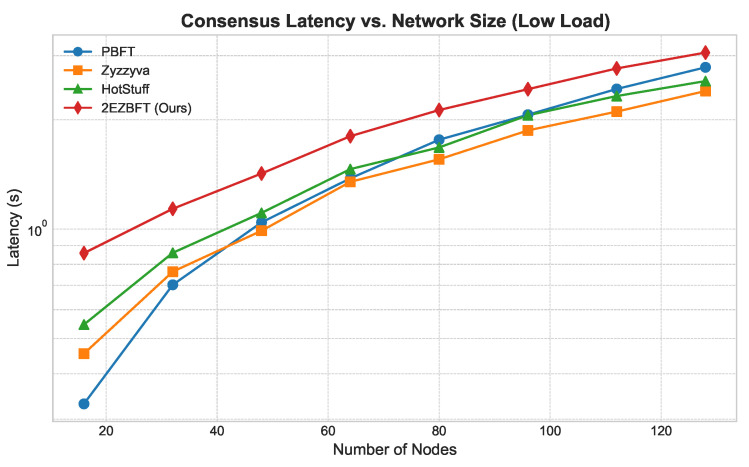
Latency vs. network size (low load).

**Figure 5 sensors-25-06268-f005:**
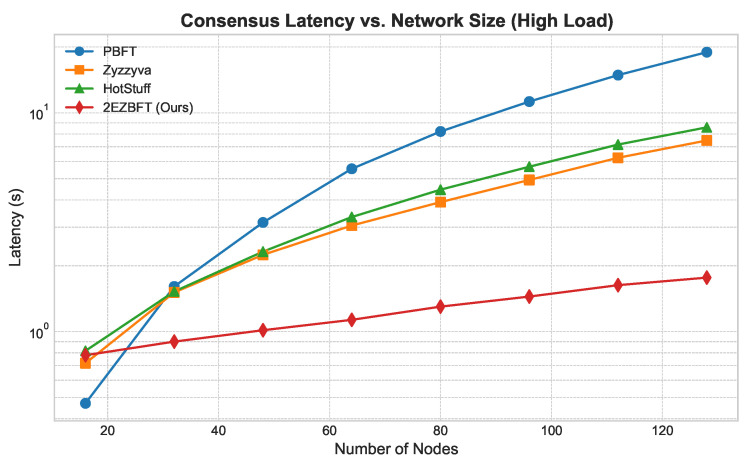
Latency vs. network size (high load).

**Figure 6 sensors-25-06268-f006:**
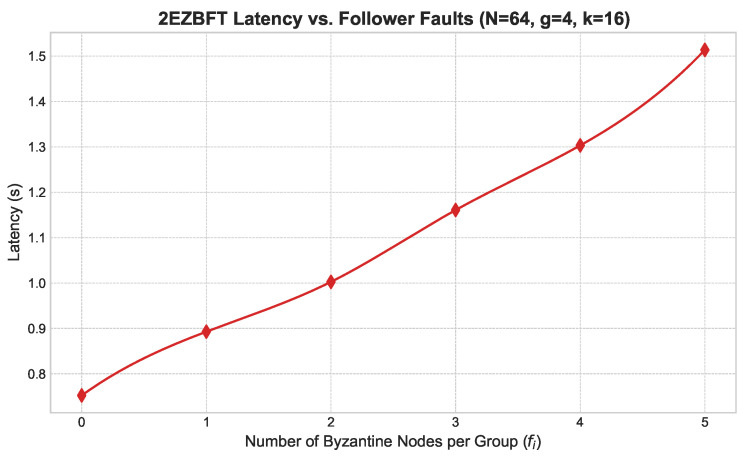
Latency vs. follower faults.

**Figure 7 sensors-25-06268-f007:**
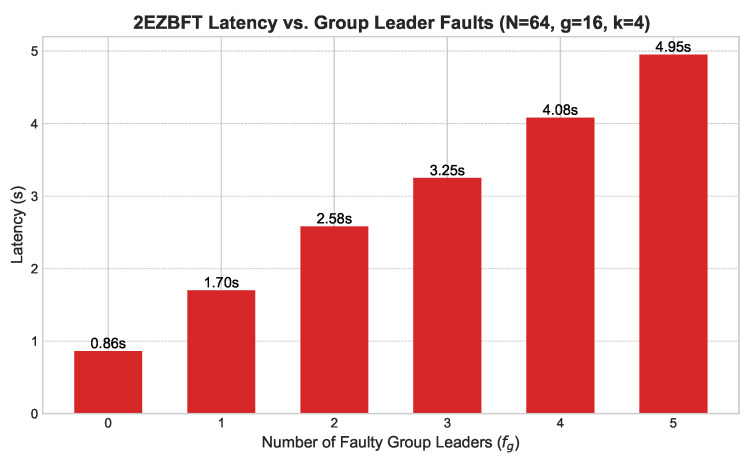
Latency vs. group leader faults.

**Figure 8 sensors-25-06268-f008:**
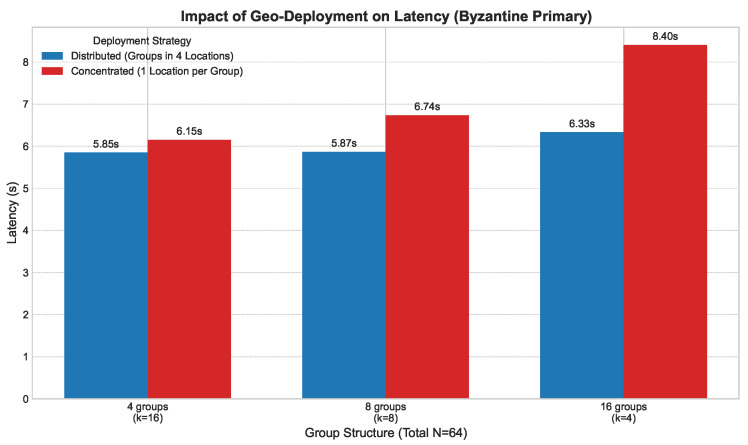
Impact of primary leader fault across group structures (*N* = 64), comparing different geographic deployment strategies.

**Figure 9 sensors-25-06268-f009:**
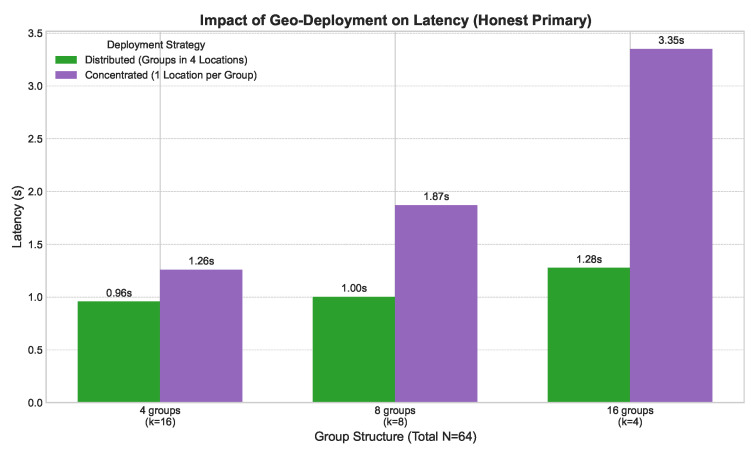
2EZBFT latency vs. group structure (*N* = 64) under different geographic deployment strategies with an honest primary leader.

**Table 1 sensors-25-06268-t001:** The symbols and variables used in the paper.

Symbol and Variable	Description
Aggregate(·)	Aggregation function (median/majority voting)
BLS(·)	Boneh–Lynn–Shacham signature scheme functions
$C_{i}$	Consensus value of group $G_{i}$
$C_{G}$	Final global consensus value
$d_{ij}$	Normalized reading of external variable collected by node $n_{ij}$ from data source
$f_{g}$	Number of Byzantine group leaders
$f_{i}$	Number of Byzantine nodes in group *i*
*g*	Number of groups
Hash(·)	Cryptographic hash function (e.g., SHA-256)
$k_{i}$	Number of nodes in group $G_{i}$
$L_{i}$	Group leader of group $G_{i}$
$L_{p}$	Primary leader of group
*N*	Total number of nodes
$n_{ij}$	Node *j* in group $G_{i}$
$P_{i}$	Proposal value computed by group leader $L_{i}$
$P_{G}$	Global proposal, aggregated from all valid group consensus values
$pk_{ij}$	Public key of node $n_{ij}$
$pk_{i}^{L}$	Public key of group leader $L_{i}$
*R*	Consensus round number, prevents replay attacks by ensuring freshness
$s_{ij}^{commit}$	Signature from node $n_{ij}$ on the commitment message (replaces $s_{ij}$)
$s_{ij}^{source}$	Source certificate signature from node $n_{ij}$ on source_id
$s_{i}^{L}$	Signature of group leader $L_{i}$ authenticating the group’s result package
$sk_{ij}$	Secret key of node $n_{ij}$
$sk_{i}^{L}$	Secret key of group leader $L_{i}$
$S_{i}^{commit}$	Aggregated commitment signature from group $G_{i}$ on $C_{i}$
$S_{i}^{source}$	Aggregated source certificate from group $G_{i}$, acting as proof of data contribution
*t*	Timestamp
*V*	Global view number (for inter-group consensus)
$V_{i}$	Group view number (for intra-group consensus in group $G_{i}$)

**Table 2 sensors-25-06268-t002:** Complexity Comparison of BFT Algorithms.

Algorithm	Message Complexity	Communication Rounds	Latency	Leader-Validator Pattern
PBFT	$O(N^{2})$	3	3Δ	Quadratic
Zyzzyva	O(N)	3 (optimistic)	3Δ	Linear
HotStuff	O(N)	3 (pipelined)	3Δ	Linear
**2EZBFT (Ours)**	** O(N) **	**3 + 4**	** 7Δ **	**Hierarchical Linear**

**Table 3 sensors-25-06268-t003:** Simulated inter-region network latency (ms).

From/To	Region A	Region B	Region C	Region D
**Region A**	5	70	90	210
**Region B**	70	5	150	120
**Region C**	90	150	5	250
**Region D**	210	120	250	5

## Data Availability

No external datasets were used, and all the references are duly cited.

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
