# Peer review of "2EZBFT for Decentralized Oracle Consensus with Distant Smart Terminals"

_sensors, 2025, doi:10.3390/s25206268_

Round 1
Reviewer 1 Report
Comments and Suggestions for Authors
The author proposes a layered BFT consensus 5 method, 2-layer EaZy BFT (2EZBFT). This method establishes multiple independent groups, each group consisting of various smart terminals, and builds "in-group synchronization, inter-group consensus", completing cross-group data aggregation and final on-chain consensus. It is innovative, but has the following problems:
1. Display the specific improvement of 2EZBFT compared to the baseline method in Abstract.
2. Strengthen the explanation of the background part and further introduce decentralized oracle.
3. Add paragraphs to the Introduction to the defects of existing methods.
4. Reference to the following latest work is missing in the related work:
Federated recommendation system based on diffusion augmentation and guided denoising
5. Increase the size of Figure 1.
6. Algorithm 1 lacks the output part, it is recommended to add the output part at the beginning.
7. The same algorithm 2 also lacks the output part.
8. Further explain the formula of 278 lines.
9. The number of existing experiments is small, and a 24-year benchmark algorithm is added to compare them to enrich the experiments.
Author Response
Comments 1: Display the specific improvement of 2EZBFT compared to the baseline method in Abstract.
Response 1: Thank you for your opinion. Abstract has been revised
Comments 2: Strengthen the explanation of the background part and further introduce decentralized oracle.
Response 2: Thank you for pointing this out. Section 2 has been revised.
Comments 3: Add paragraphs to the Introduction to the defects of existing methods.
Response 3: Thank you for pointing this out. Section 2 has been revised.
Comments 4: Reference to the following latest work is missing in the related work: Federated recommendation system based on diffusion augmentation and guided denoising
Response 4: Thank you for pointing this out. Section 2 has been revised.
Comments 5: Increase the size of Figure 1.
Response 5: Thank you for pointing this out. The Figure has been revised
Comments 6: Algorithm 1 lacks the output part, it is recommended to add the output part at the beginning.
Response 6: Thank you for pointing this out. The Outputs have been added.
Comments 7: The same algorithm 2 also lacks the output part.
Response 7: The Outputs have been added.
Comments 8: Further explain the formula of 278 lines.
Response 8: The formula is assumed for delay analysis, which have been explained in paragraph ahead.
Comments 9: The number of existing experiments is small, and a 24-year benchmark algorithm is added to compare them to enrich the experiments.
Response 9: Thank you for the suggestion. Our study focuses on WAN congestion under flat broadcast patterns; hence we selected PBFT, Zyzzyva, and HotStuff as representative baselines spanning quadratic, optimistic, and linear regimes. Many recent protocols (e.g., committee/DAG-backed) optimize along orthogonal axes (stake-based sampling, asynchronous RBC, or decoupled mempools), which would confound a WAN-queuing–centric comparison without substantial reimplementation.
Reviewer 2 Report
Comments and Suggestions for Authors
Summary:
The authors propose a two-layer Byzantine Fault Tolerant (BFT) consensus protocol tailored for decentralized oracle networks with geographically distributed smart terminals.
Overall, the topic is interesting and a field of active research, since it is crucial that the real-world data utilized as triggering-events in smart contracts should be of high validity.
Comments:
In my opinion, the architecture proposed by the authors is a “general-purpose” scalable-consensus mechanism, (not explicit or specific to Oracles).
To the best of my efforts I found hard to understand whether the aggregating nodes collect data from nodes/oracles observing the same or different physical-world events (i.e. the same or different real-word data sources).
To the best I can tell, in this work the validity of intra-group aggregation seems to implicitly rely on the assumption that all nodes observe the same external variable. If this holds, it should be noted explicitly, and its limitations should be discussed.
In PBFT with oral messages a general requirement for f<N/3 where f is the number of Byzantine nodes is posed. Are the values of “maximum number of faults tolerated within a single group” relevant here?
An overall descriptive architectural diagram of the solution would be helpful to the reader should be included by the authors.
Author Response
Comments 1: In my opinion, the architecture proposed by the authors is a “general-purpose” scalable-consensus mechanism, (not explicit or specific to Oracles).
Response 1: We appreciate this insight. Our design indeed reuses general BFT building blocks (hierarchical partitioning + signature aggregation) and could be extended more broadly. However, the current work cannot satisfy the consensus in general message situations. So we intentionally specialize these mechanisms for geo-distributed oracle data collection.
Comments 2: To the best of my efforts I found hard to understand whether the aggregating nodes collect data from nodes/oracles observing the same or different physical-world events (i.e. the same or different real-word data sources)
Response 2: All nodes participating in that round observe or measure the same real‑world metric, but from different sources. We revised the description about this part in section 3.
Comments 3: To the best I can tell, in this work the validity of intra-group aggregation seems to implicitly rely on the assumption that all nodes observe the same external variable. If this holds, it should be noted explicitly, and its limitations should be discussed.
Response 3: Thank you for pointing this out. We have noted it explicitly and added discussion about this part in Section 4.3
Comments 4: In PBFT with oral messages a general requirement for f<N/3 where f is the number of Byzantine nodes is posed. Are the values of “maximum number of faults tolerated within a single group” relevant here?
Response 4: As shown in Table of the symbols and variables used in the paper, f_i represents the number of Byzantine group leaders and k_i is the number of Byzantine nodes in group i. We discuss the faulty node separately because of the limitation of layered network structure.
Comments 5: An overall descriptive architectural diagram of the solution would be helpful to the reader should be included by the authors.
Response 5: Thank you for your opinion. We added a picture of the architecture in Section 3.
Reviewer 3 Report
Comments and Suggestions for Authors
This paper deals with the traditional consensus mechanisms problems in geo-distributed sensor environments. The authors propose a layered BFT consensus method, named 2-layer EaZy BFT(2EZBFT). The layered design reduces the complexity of intra-group communication and reduces the number of cross-group interactions, thereby lowering the overall network overhead. The paper is well-written and well-structured.
Before publishing, authors should improve the paper in accordance with the following comments:
- The authors stated that one of their contributions is an extensive evaluation against established BFT protocols, such as PBFT, Zyzzyva, HotStuff, Tendermint/CometBFT, and HoneyBadgerBFT. Authors should explain why they omitted BFT (Scalable BFT), RBFT (Redundant BFT), Mir-BFT (by Facebook/Meta), Algorand, and EZBFT, etc. The criteria for this selection should be explained.
- Later in the paper, the proposed method is compared only with PBFT, Zyzzyva, and HotStuff, so the absence of the other mentioned protocols in the introduction should be explained.
- The whole Related Works section is too short. There is no critical overview of the presented methods. Also, there is no clear definition of what their proposal solves as a problem compared to existing solutions.
- The paper should be expanded with a detailed description of the simulation written in Go.
- Authors should explain better the different behaviors of the latency shown in Fig. 3. and Fig. 4 (Low and High load scenarios),
- The Conclusion section should be expanded with a summarization of results, and with constraints and limitations of the study if possible. The motivation for the future development should be explained as well.
Author Response
Comments 1: The authors stated that one of their contributions is an extensive evaluation against established BFT protocols, such as PBFT, Zyzzyva, HotStuff, Tendermint/CometBFT, and HoneyBadgerBFT. Authors should explain why they omitted BFT (Scalable BFT), RBFT (Redundant BFT), Mir-BFT (by Facebook/Meta), Algorand, and EZBFT, etc. The criteria for this selection should be explained.
Response 1: Thank you for pointing this out. We have added the explanation in Section 5
Comments 2: Later in the paper, the proposed method is compared only with PBFT, Zyzzyva, and HotStuff, so the absence of the other mentioned protocols in the introduction should be explained.
Response 2: Thank you for pointing this out. We have added the explanation in Section 5.
Comments 3: The whole Related Works section is too short. There is no critical overview of the presented methods. Also, there is no clear definition of what their proposal solves as a problem compared to existing solutions.
Response 3: Thank you for pointing this out. Section 2 has been revised.
Comments 4: The paper should be expanded with a detailed description of the simulation written in Go.
Response 4: We have added Section 5.2 to describe the simulation design.
Comments 5: Authors should explain better the different behaviors of the latency shown in Fig. 3. and Fig. 4 (Low and High load scenarios),
Response 5: We appreciate this insight. We have revised the description.
Comments 6: The Conclusion section should be expanded with a summarization of results, and with constraints and limitations of the study if possible. The motivation for the future development should be explained as well.
Response 6: We appreciate this insight. And we have revised Section 5.
Round 2
Reviewer 2 Report
Comments and Suggestions for Authors
The authors have addressed most of my comments comprehensively.
I suggest the paper to be accepted for publication in its current form.
Author Response
Thank you. We have revised Figures 3, 4, and 5 to include markers in the plot legends.